# ARMCX3 Mediates Susceptibility to Hepatic Tumorigenesis Promoted by Dietary Lipotoxicity

**DOI:** 10.3390/cancers13051110

**Published:** 2021-03-05

**Authors:** Serena Mirra, Aleix Gavaldà-Navarro, Yasmina Manso, Mónica Higuera, Román Serrat, María Teresa Salcedo, Ferran Burgaya, José Maria Balibrea, Eva Santamaría, Iker Uriarte, Carmen Berasain, Matias A. Avila, Beatriz Mínguez, Eduardo Soriano, Francesc Villarroya

**Affiliations:** 1Department of Cell Biology, Physiology and Immunology, University of Barcelona, 08007 Barcelona, Spain; serena.mirra@ub.edu (S.M.); ymansosanz@ub.edu (Y.M.); roman.serrat@inrae.fr (R.S.); fburgaya@ub.edu (F.B.); 2Centro de Investigación Biomédica en Red de Enfermedades Neurodegenerativas (CIBERNED), Instituto de Salud Carlos III, 28029 Madrid, Spain; 3Department of Biochemistry and Molecular Biomedicine and Institute of Biomedicine, University of Barcelona, 08028 Barcelona, Spain; aleixgavalda@ub.edu; 4Centro de Investigación Biomédica en Red Fisiopatología de la Obesidad y Nutrición (CIBEROBN), Instituto de Salud Carlos III, 28029 Madrid, Spain; 5Liver Diseases Research Group, Vall d’Hebron Institute of Research, VHIR, 08035 Barcelona, Spain; monica.higuera@vhir.org(M.H.); bminguez@vhebron.net(B.M.); 6Pathology Department, Hospital Universitari Vall d’Hebron, Universitat Autònoma de Barcelona, 08035 Barcelona, Spain; mtsalcedo@vhebron.net; 7Endocrine, Metabolic and Bariatric Surgery Unit, General Surgery Department, Hospital Universitari Vall d’Hebron, 08035 Barcelona, Spain; balibrea@clinic.cat; 8Centro de Investigación Biomédica en Red de Enfermedades Hepáticas y Digestivas (CIBEREHD), Instituto de Salud Carlos III, 28029 Madrid, Spain; evasmaria@unav.es(E.S.);iuriarte@unav.es(I.U.); cberasain@unav.es(C.B.); maavila@unav.es(M.A.A.); 9Hepatology Programme, CIMA-University of Navarra, IdiSNA, 31009 Pamplona, Spain; 10Liver Unit, Department of Internal Medicine, Hospital Universitari Vall d’Hebron, Universitat Autònoma de Barcelona, 08035 Barcelona, Spain

**Keywords:** Alex3, HCC, lipotoxicity, NAFLD, obesity, SOX9

## Abstract

**Simple Summary:**

An excess fat in the liver enhances the susceptibility to hepatic cancer. We found that Armcx3, a protein only known to date to play a role in neural development, is strongly increased in mouse liver in response to lipid availability and proliferation-inducing insults. In patients, the levels of hepatic Armcx3 are also increased in conditions of high exposure of the liver to fat. We wanted to determine the role of Armcx3 in the hepatocarcinogenesis favored by a high-fat diet. We generated mice with genetically driven suppression of Armcx3, and we found that they were protected against experimentally induced hepatic cancer, especially in conditions of a high-fat diet. Armcx3 was also found to promote hepatic cell proliferation through the interaction with Sox9, a known proliferation factor in hepatocellular carcinoma. Armcx3 is identified as a novel factor in meditating propensity to liver cancer in conditions of high hepatic lipid insults.

**Abstract:**

ARMCX3 is encoded by a member of the Armcx gene family and is known to be involved in nervous system development and function. We found that ARMCX3 is markedly upregulated in mouse liver in response to high lipid availability, and that hepatic ARMCX3 is upregulated in patients with NAFLD and hepatocellular carcinoma (HCC). Mice were subjected to ARMCX3 invalidation (inducible ARMCX3 knockout) and then exposed to a high-fat diet and diethylnitrosamine-induced hepatocarcinogenesis. The effects of experimental ARMCX3 knockdown or overexpression in HCC cell lines were also analyzed. ARMCX3 invalidation protected mice against high-fat-diet-induced NAFLD and chemically induced hepatocarcinogenesis. ARMCX3 invalidation promoted apoptotic cell death and macrophage infiltration in livers of diethylnitrosamine-treated mice maintained on a high-fat diet. ARMCX3 downregulation reduced the viability, clonality and migration of HCC cell lines, whereas ARMCX3 overexpression caused the reciprocal effects. SOX9 was found to mediate the effects of ARMCX3 in hepatic cells, with the SOX9 interaction required for the effects of ARMCX3 on hepatic cell proliferation. In conclusion, ARMCX3 is identified as a novel molecular actor in liver physiopathology and carcinogenesis. ARMCX3 downregulation appears to protect against hepatocarcinogenesis, especially under conditions of high dietary lipid-mediated hepatic insult.

## 1. Introduction

The Armadillo repeat-containing X-linked (Armcx) gene cluster encodes proteins that contain multiple Armadillo domains. The functional role of distinct Armcx family members is still poorly known. Recently, the ARMCX3 member of the family has been reported to show a high neuronal expression and to be involved in the regulation of mitochondrial dynamics in neurons [1,2]. Previous research also showed that ARMCX3 suppresses canonical Wnt signaling and controls the proliferation of neural precursor cells during chicken spinal cord development [3]. On the other hand, *ARMCX1-3* are downregulated in a number of human carcinomas [4] and recent studies have indicated that members of the Armcx family are involved in overall growth, tumorigenesis and tumor cell invasion [5,6,7,8,9,10]. Thus, ARMCX1 over-expression has been reported to suppress colony formation in colorectal carcinoma cell lines [7] whereas ARMCX3 has also been proposed to suppress non-small cell lung cancer invasion through the AKT/Slug/E-cadherin signaling pathway [9]. Conversely, *ARMC10*, which is a close phylogenetically related gene [1], has been found to be upregulated in hepatocellular carcinoma (HCC) [11].

Nonalcoholic fatty liver disease (NAFLD), which is associated with excessive accumulation of fat (triglycerides) in the liver in the absence of excessive alcohol consumption, is the most common chronic liver disease worldwide [12]. The severe form of NAFLD, nonalcoholic steatohepatitis (NASH), is associated with inflammation and hepatic injury and can progress to liver cirrhosis and HCC [13]. Obesity increases the risk for NASH and hepatocarcinogenesis [14,15]. Lipid accumulation in hepatocytes exacerbates inflammatory processes and generates hepatic chronic stress, leading to the functional alteration of intracellular organelles, such as mitochondria and the endoplasmic reticulum [16]. However, the underlying cellular mechanisms and molecular actors linking NAFLD with hepatocarcinogenesis are not fully known. We found that ARMCX3 is highly expressed in liver of patients with HCC whereas hepatic ARMCX3 expression in mice is highly responsive to dietary challenges, mostly those leading to fat accumulation in liver. In light of these observations, we undertook the analysis of the role of ARMCX3 in liver pathophysiology in relation to hepatic lipotoxicity and carcinogenes using *Armcx3* inducible knockout mice in vivo, and loss- and gain-of-function models in vitro. We found that ARMCX3 promotes the proliferative processes of hepatic cells and enhances hepatocarcinogenic susceptibility, especially in response to lipotoxicity-meditated insults.

## 2. Results

### 2.1. Hepatic Expression of Armcx3 Is Modulated in Response to Nutritional Challenges

*Armcx3* has been mainly studied in the central nervous system [1,2,3], but it is also expressed in several other tissues, including the liver, muscle and white adipose tissue (Appendix A). We found that the expression of *Armcx3* in liver is strongly modulated according to variations in the nutritional status of mice (Figure 1A): mice fasted overnight showed a dramatic reduction in the hepatic expression of *Armcx3*, whereas those fed a high-fat diet (HFD) showed significantly higher levels of *Armcx3* mRNA and ARMCX3 protein in the liver. Among the plethora of metabolic effects linked to HFD, NAFLD is a severe consequence. Acute treatment of mice with tunicamycin, an experimental model for the short-term induction of hepatosteatosis and NAFLD [17], caused significant upregulation of *Armcx3* mRNA and ARMCX3 protein in liver. Concordant with these data obtained in mice, we also found increased *ARMCX3* mRNA expression in liver samples from a cohort of patients with NASH showing overt NAFLD, relative to controls (Figure 1B).

### 2.2. Gene Invalidation of Armcx3 Ameliorates HFD-Induced Metabolic Alterations and Protects against NAFLD

To explore the possible function of ARMCX3 in metabolic regulation, especially in the liver, we generated an *Armcx3* inducible knockout mouse (*f*Armcx3/Cre) by crossing *Armcx3* floxed (*f*Armcx3) mice with tamoxifen-inducible UBC-Cre/ERT2 recombinase (Cre) mice (Appendix A). Mating *f*Armcx3 mice with *f*Armcx3/Cre+ mice allowed us to obtain from the same litter two experimental groups: *f*Armcx3/Cre- (Control) and *f*Armcx3/Cre+ (ARMCX3-KO). *f*Armcx3/Cre+ pups and *f*Armcx3/Cre- controls were injected with tamoxifen to induce CRE activity. ARMCX3 protein levels were practically undetectable in liver and other tissues of tamoxifen-injected ARMCX3-KO mice by day 15 of life (Appendix A). ARMCX3-KO mice developed normally and did not display any obvious behavioral defect. ARMCX3 invalidation was maintained in adults, as determined at the end point of the experimental procedure in adulthood.

Six-week-old control and ARMCX3-KO mice were fed either standard, low-fat diet (LFD) or HFD for 16 weeks (Figure 1C). Control and ARMCX3-KO mice showed similar weight gains when fed LFD, and HFD exposure triggered increased body weight, as expected. Notably, ARMCX3-KO mice fed HFD gained less body weight than controls fed HFD, even though the ARMCX3-KO mice had a higher food intake. Basal glycemia was not significantly altered in ARMCX3-KO mice (Figure 1D), whereas, under HFD, glucose tolerance was better in ARMCX3-KO mice relative to controls (Figure 1E). Insulin levels were also higher in ARMCX3-KO mice, especially in the LFD-fed group (Figure 1D). The HOMA-IR index was increased in HFD-treated controls, as expected, but this increase was blunted in HFD-fed ARMCX3-KO mice. Thus, invalidation of *Armcx3* appeared to protect against HFD-induced glucose homeostasis impairment. Histological analysis revealed that ARMCX3-KO mice were resistant to developing HFD-induced NAFLD (Figure 1F). Moreover, the protection against NAFLD in ARMCX3-KO mice was accompanied by less HFD-induced alanine aminotransferase (ALT) activity in plasma (Figure 1G) indicating reduced hepatocellular injury [18]. However, not all liver function tests (LFTs) evidenced marked effects, despite a non-significant trend of lower activities of alkaline phosphatase (ALP), aspartate amino transferase (AST) and lactate dehydrogenase (LDH) in ARMCX3-KO mice, especially under HFD (Appendix A). Together, these results indicate that depletion of ARMCX3 protects mice against HFD-induced metabolic insult, improves glucose homeostasis and reduces to some extent NAFLD-related liver injury.

### 2.3. Inactivation of the Armcx3 Gene Protects Against Hepatic Carcinogenesis

Long-term HFD feeding favors the development of NAFLD, which can progress to fibrosis, cirrhosis and even hepatocarcinoma (HCC) [19]. Moreover, experimental HFD has been described to enhance the development of chemically induced HCC [20]. In a cohort of 48 patients with HCC, we found that 17 (35%) of the patients had hepatic *ARMCX3* mRNA levels at least 2.5-fold higher than those of control subjects. Thus, we focused our study on the role of ARMCX3 in liver carcinogenesis and its interaction with lipotoxicity-related insults. Pups originating from *f*A3x*f*A3/Cre+ crossings were treated with tamoxifen on days 1 to 3 after birth to induce *Armcx3* invalidation, and 14 days later the mice were injected with diethylnitrosamine (DEN) (25 mg/kg) to induce hepatic carcinogenesis (Figure 2A). At 4 weeks after DEN administration, control and ARMCX3-KO male mice were fed either LFD or HFD.

Similar to what was shown previously in the absence of DEN, DEN-injected ARMCX3-KO mice had a lower increase in body weight compared to DEN-injected control mice, regardless of diet (Figure 2B). This was due to reduced adiposity, as reflected by smaller adipose depots (Appendix A), whereas there was no apparent difference in mouse growth (as assessed by tibia length). ARMCX3-KO mice fed LFD also showed lower glycemia (Appendix A).

Mice were evaluated for liver tumors development at 30 weeks of age. Representative images of the livers are shown in Figure 2C. DEN-injected control mice fed HFD exhibited many more tumors per liver than mice fed LFD, as previously reported [21]. The tumor size and maximum tumor size in DEN-treated mice were also higher in HFD-fed mice compared to LFD-fed mice. We observed that DEN treatment of ARMCX3-KO mice resulted in decreases in the tumor number, average tumor size and maximum tumor size compared with those of control littermates, indicating that ARMCX3 deficiency had a marked and significant protective effect in HFD-fed mice.

Histological analyses revealed that DEN-treated control mice fed HFD exhibited marked hepatosteatosis with abundant intracellular lipid droplets (Figure 2D,E). We also observed extended and widespread tumoral foci that resulted in a deep alteration of the hepatic histological organization, which was consistent with the enhanced number of tumors. Hepatic steatosis was strongly reduced in the livers of ARMCX3-KO mice maintained on HFD. Similar to our findings for ARMCX3-KO mice in the absence of DEN, HFD increased ALT activity in DEN-treated control mice, and this induction was blunted in DEN-treated ARMCX3-KO mice (Figure 2F), whereas a minor impact of ARMCX3 invalidation was detected for other LFTs (Appendix A).

Overall, these results indicate that the expression of ARMCX3 determines the susceptibility of mice to DEN-induced tumorigenesis and its enhancement in response to the hepatic metabolic insults elicited by an HFD. Conversely, impaired ARMCX3 expression protects against the hepatic tumorigenesis and metabolic derangements elicited by HFD.

### 2.4. ARMCX3 Regulates DEN-Induced Apoptotic Death and Macrophage Infiltration 

Proliferation and apoptosis are key events in tumor evolution and progression. In addition to being a carcinogenic agent per se, DEN is known to cause accumulation of reactive oxygen species resulting in hepatocytes cell death, which triggers apoptosis-induced proliferation as a compensatory event [22,23]. Cellular proliferation was analyzed by Ki67 staining and in vivo BrdU labeling, although only a limited number of tumors was available from ARMCX3-KO mice given the protective effect of ARMCX3 deficiency. Our results indicated that ARMCX3 depletion did not affect markedly cellular proliferation in tumor regions, although Ki67 and BrdU positive staining tended to be higher in ARMCX3-KO mice that were fed HFD (Figure 3A,B). Cleaved caspase-3 was assessed as an indicator of apoptotic activity in hepatic non-tumor and tumor regions. ARMCX3 invalidation did not affect apoptosis of hepatocytes in non-tumor regions, but tumor regions from ARMCX3-KO mice fed HFD showed markedly more cleaved caspase-3 relative to those of control mice (Figure 3C). Concordant with these apoptosis-related data, immunostaining of F4/80 cells indicated that there was greater accumulation of macrophages in tumors from HFD-fed ARMCX3-KO mice compared to HFD-fed control mice and LFD-fed ARMCX3-KO mice (Figure 3D). These results indicate that the lack of ARMCX3 promotes hepatocellular apoptosis and macrophage infiltration in tumors from DEN-treated mice maintained on HFD.

### 2.5. ARMCX3 Invalidation Does Not Affect Hepatic Mitochondria

ARMCX3 has been detected in mitochondria, nuclei and cytosol [1,24], and the mitochondrial localization of ARMCX3 has been described as being crucial for regulating the function of ARMCX3 in the brain [1,3,24]. Immunohistochemical analysis of the mitochondrial marker cytochrome-c oxidase subunit IV (COX-IV) did not reveal any alteration in the distribution of mitochondria in ARMCX3-KO and control hepatocytes from LFD- or HFD-fed mice (Appendix A). Similarly, transmission electron microscopy analysis of hepatocytes did not reveal any remarkable change in the morphology, abundance or size of mitochondria due to ARMCX3 invalidation or dietary condition (Appendix A).

### 2.6. ARMCX3 Affects ERK Signaling in DEN-Induced Tumors from HFD-Fed Mice

ARMCX3 overexpression downregulates the Wnt/β-catenin pathway in neuronal cell [3]. β-catenin signaling is a major oncogenic pathway in HCC, involved in both anti-apoptotic and proliferative processes [25]. We observed an intense immunoreactivity of β-catenin in hepatic tumors from the control and ARMCX3-KO groups, as expected, but similar low levels of nuclear β-catenin accumulation (Figure 4A). These data do not support a relevant involvement of the β-catenin pathway in determining the differential behavior of ARMCX3-KO mice in relation to tumor development, proliferation and apoptosis in hepatic cells.

Key intracellular regulatory pathways, such as the p38 MAPK and ERK pathways, have relevant effects on hepatic proliferation and carcinogenesis [26,27] and are also involved in hepatic energy and lipid metabolism [28,29,30]. We examined the impact of ARMCX3 invalidation on the phospho-AKT/AKT, phospho-S6/S6, phospho-ERK/ERK and phospho-p38/p38 ratios in non-tumor tissue from DEN-treated LFD and HFD-fed mice. No significant effects were found due to genotype or diet-genotype interaction (data not shown). We found that ERK activation was significantly higher in tumors of control mice (Figure 4B), consistently with a previous report [20]. This difference was not found in ARMCX3-KO mice. p38 MAPK activation tended to be lower in tumor tissues, but irrespective of invalidation or not of ARMCX3. Thus, among the tested pathways, only ERK appeared to be specifically affected in liver due to ARMCX3 invalidation.

### 2.7. ARMCX3 Invalidation Reduces Cell Viability, Clonality and Migration in Hepatocellular Carcinoma Cell Lines

To assess whether the protective effects of ARMCX3 invalidation on hepatic tumorigenesis in mice may occur in a cell-autonomous manner, we performed siRNA-mediated knockdown of *ARMCX3* (80–90% of reduction of ARMCX3 expression) in two human HCC cell lines with high basal expression of ARMCX3: SNU423 and HepB3 (Figure 5A). We found a significant reduction in the clonality capacity of ARMCX3-silenced SNU423 and Hep3B cells relative to controls (Figure 5B). Moreover, the siRNA-mediated impairment of ARMCX3 expression significantly reduced the migration and potential invasiveness of SNU423 HCC cells (Figure 5C) as assessed in wound-healing assays. We also found that *ARMCX3* knockdown resulted in a marked and significant reduction in cell viability, measured by MTT, in both HCC cell lines (Figure 5D). 

Together, these data support the notion that ARMCX3 downregulation, in addition to having potential protective effects against hepatic tumorigenesis mediated by amelioration of the metabolic profile and subsequently reduction of lipotoxicity in the liver, also exerts direct and cell-autonomous effects consistent with an anti-proliferative action on hepatic cells.

### 2.8. ARMCX3 Overexpression In Vitro Induces Hepatocellular Carcinoma Cells Proliferation in a SOX9-Dependent Mechanism 

To complement our observations that ARMCX3 silencing reduces cell viability and clonality, we analyzed these parameters in ARMCX3-overexpressing SNU423 cells (Figure 6A). Adenoviral vector-mediated overexpression of ARMCX3 led to a significant increase in cell proliferation compared to Ad-GFP-transduced cells. In addition, overexpression of ARMCX3 promoted the clonality capacity of hepatoma cells, resulting in an increase in the size of colonies. Overexpression of ARMCX3 increased the protein level of PCNA, a well-established marker of cell proliferation, in SNU423 cells (Figure 6B) and even in mouse primary hepatocytes (Figure 6C), thus confirming the proliferation-inducing effect triggered by the overexpression of ARMCX3. We explored the effects of ARMCX3 over-expression on the intracellular pathways previously analyzed in the “in vivo” invalidation experiments above. We found a significant increase in P-ERK/ERK ratio in response to increased ARMCX3, reciprocally to the reduction found in liver from ARMCX3-KO null mice, which supported the involvement of the ERK pathway in the ARMCX3 intracellular actions. Over-expression of ARMCX3 also induced the relative phosphorylation of beta-catenin and trended to increase p38 MAP kinase phosphorylation (Appendix A).

Given our observations that overexpression of ARMCX3 induces proliferation, we questioned whether ARMCX3 expression would be regulated in experimental models of hepatocyte proliferation in vivo, such as partial hepatectomy and CCl_4_ treatment (Figure 6D) [31]. Interestingly, the hepatic ARMCX3 protein levels were significantly induced at 48 h after the experimental removal of 66% of the liver. Treatment of mice with CCl_4_, which is known to lead to acute hepatic injury followed by induction of proliferation in order to regenerate hepatic tissue, induced the expression of ARMCX3 concordantly with induction of PCNA protein levels. Analysis of the relative inductions of PCNA and ARMCX3 in these experimental models revealed that these parameters exhibited a very strong and significant positive correlation (Figure 6E). Thus, hepatocyte proliferation is highly associated with enhanced ARMCX3 expression. Given the relationship we observed between ARMCX3 and hepatic lipid metabolism in vivo (Figure 1), we determined the effects of fatty acids on ARMCX3 expression in hepatoma cells. Indeed, palmitic acid induced *ARMCX3* mRNA levels in HepG2 cells, whereas oleate (a much less lipotoxic fatty acid) did not (Figure 6F).

Members of the ARMC protein family reportedly interact with members of the sex-determining region Y-box (SRY-box) containing transcription factor (SOX) family, whose members are transcription factors that participate in multiple processes ranging from development and cell differentiation to tumorigenesis. Specifically, ARMCX3 was shown to interact with SOX10 [24], while SOX9 is reportedly overexpressed in HCC [32]. We found, using co-immunoprecipitation assays, that ARMCX3 and SOX9 physically interact in hepatoma cells (Figure 7A). We then performed siRNA-mediated knockdown of SOX9 in HepG2 cells previously induced to overexpress ARMCX3. We found that the proliferative effects of ARMCX3 overexpression were reduced upon silencing of SOX9, as indicated by the downregulation of PCNA (Figure 7B). In fact, ARMCX3 overexpression itself increased SOX9 protein levels. In light of these data using in vitro models, we determined whether SOX9 levels were altered in the liver from ARMCX3-KO mice. We found that the invalidation of ARMCX3 resulted in a concomitant reduction of hepatic SOX9 levels, especially under HFD conditions (Figure 7C). These data collectively show the regulatory interdependence between ARMCX3 and SOX9, and support the notion that ARMCX3 may promote hepatocarcinogenesis through pathways involving SOX9. 

## 3. Discussion

In the present study, we report that ARMCX3 is expressed in the liver and is markedly upregulated under conditions of high hepatic exposure to lipids or enhanced hepatic cell proliferation. Experimentally induced ARMCX3 downregulation exerted protective effects against the metabolic derangements elicited by an HFD and the development of chemically induced hepatic tumors, especially in the presence of HFD-mediated insults.

Metabolic syndrome and liver steatosis are associated with NAFLD [33,34,35], which may be an initial stage in the development of NASH and HCC [36]. The protective effects of ARMCX3 downregulation on systemic metabolism and hepatic steatosis, especially under HFD, may be involved in the ability of ARMCX3 downregulation to protect against hepatocarcinogenesis. The high levels of hepatic ARMCX3 found in conditions of NAFLD (both experimentally induced and in patients) and the induction of ARMCX3 seen in response to palmitate treatment of hepatic cells may be part of the deleterious effects seen when liver is exposed to excessive fat and the subsequent hepatic lipotoxicity, which primes for potential HCC. 

Our data suggest that ARMCX3 plays an additional cell-autonomous role in hepatic cells that would be consistent with the susceptibility to HCC. We found that ERK phosphorylation and especially the interaction with SOX9 are involved in the effects of ARMCX3. We report that SOX9 interacts with ARMCX3, which is to be added to a previous observation of interaction with SOX10 [24], and that ARMCX3 and SOX9 reciprocally regulate each other’s protein levels. HCC progenitor cells, which are pre-malignant actors that give rise to cancer when the liver is under chronic damage and compensatory proliferation has been activated, express *SOX9* [37]. Moreover, *SOX9* expression has been correlated with liver-damage-associated fibrosis, and cells expressing *SOX9* have been found to expand in early human hepatocarcinogenesis [38]. It may be hypothesized that the pro-tumorigenic activity of ARMCX3 may occur during the first stages of hepatocyte transformation, and that this is supported by a combination of interaction with SOX9 and a damaging environment elicited by fat overload. We did not find any difference in cell proliferation (Ki67 and BrdU staining) between control and ARMCX3-KO mice in tumor regions, but such differences were evident in ARMCX3-KO non-tumor areas. This supports the notion that the pro-tumorigenic activity of ARMCX3 may occur during the first stages of hepatocyte transformation rather than in cells that are already tumoral.

Our in vitro results do not allow us to rule out the possibility that ARMCX3 controls tumor progression by enhancing proliferation and invasiveness. Silencing of ARMCX3 protein reduced the in vitro proliferation rate in a manner consistent with a positive role of ARMCX3 on cell proliferation, clonality capacity and migration in HCC cells. Conversely, ARMCX3 overexpression promoted clonality and proliferation, even in primary hepatocytes, which are characterized by a very poor proliferation potential in culture [39]. Therefore, ARMCX3 increases the proliferative capacity of hepatocytes, which is fully consistent with the increased ARMCX3 protein levels seen in liver that is regenerating after experimental hepatic insult.

We propose that lipotoxicity may favor the tumorigenic potential of ARMCX3 and the involvement of ERK and SOX9. This is consistent with previous reports that palmitate treatment enhances ERK phosphorylation [40] and induces *Sox9* [41], and that ERK phosphorylation increases SOX9 [42]. In summary, in contrast with renal and lung in vitro systems in which ARMCX3 has been reported to protects against carcinogenic events [9], our findings highlight a distinct action of ARMCX3 as promoting tumorigenesis in liver, especially in the context of lipotoxicity, what is in line with previous observations for the related *Armc10* gene product in hepatic cells [11]. It is possible that our findings in hepatic cells, reinforced by our in vivo in mouse models, point to a tissue-specific distinct role of ARMCX3 (and, perhaps, other Armcx family members) in relation to tumorigenesis, which warrants further research. 

## 4. Materials and Methods

### 4.1. Mice Care

Mice were bred in the animal research facilities at the University of Barcelona. Animals were provided with food and water ad libitum and maintained in a temperature-controlled environment in a 12/12 h light-dark cycle. 

### 4.2. Human Samples

Samples were collected prospectively from patients treated by bariatric surgery as a steatohepatic liver. Tissue from control patients treated by surgical resection of metastasic lesions in liver from a colon adenocarcinoma were obtained from the Tissue Bank at Vall d’Hebron Hospital. Clinical characteristics are described in Appendix A for NASH group and in Appendix A for Control group.

Liver biopsies were evaluated by an expert liver pathologist. NASH was diagnosed applying the NASH CRN score which describes the nonalcoholic fatty liver disease activity score (NAS), which is a composite score of steatosis, lobular inflammation, cytological ballooning and fibrosis (disease stage). NAS score is defined as the unweighted sum of scores for steatosis (0−3), lobular inflammation (0−3) and ballooning (0−2), ranging from 0 to 8 [43]. Liver fibrosis was diagnosed by the Metavir scale (defined as: F0, no fibrosis; F1, enlarged portal tract without septa; F2, enlarged portal tract with rare septa or bridging fibrosis; F3, bridging fibrosis without cirrhosis; F4, cirrhosis) [44].

### 4.3. ARMCX3-KO Mouse Generation

The ARMCX3-knock-out targeting construct was designed and constructed at the IRB Mutant Mouse facility using standard recombineering methods [45]. The targeting vector was confirmed by sequencing and linearized vector transfected into ES cells for targeting. Cells were selected with neomycin and genomic DNA analyzed by PCR and by Southern blot for correct targeting. PCR analysis was used for screening the 5′ and 3′ insertion sites of the ARMCX3 knock-out allele. Digested genomic DNA was analyzed following SacI digestion by southern analysis and correctly targeted clones were injected into C57B6/J blastocysts and reimplanted into the oviduct of 2.5-day pseudopregnant foster mice (CD1s). Chimeras were mated with heterozygous Flp mice to remove the neomycin selection marker and to obtain the floxed-ARMCX3 mouse (fARMCX3). fARMCX3 was mated to a UBC-Cre/ERT2 mouse (The Jackson Laboratory, Bar Harbor, ME) to obtain control and fARMCX3/Cre+ (CKO, conditional knock-out mouse). Activation of the inducible Cre recombinase was achieved by daily intra-gastric tamoxifen injection of pups at postnatal days 1 to 3 (P1–P3) to obtain ARMCX3-KO mice.

Control and ARMCX3-KO male mice were selected for the study and genotyped using the following primers for fARMCX3:*Armcx3 S1F*: GGGGCGGTGGGCAGGATGACAGC*Armcx3 S4F*: AAGTTCTAGGAATCGAGAGCC*Armcx3 S:* ATCATTTCCCCTTGACTCTGG 

And the following primers for CRE:*Forward Wt-Cre*: CTAGGCCACGAATTGAAAGATCT*Reverse Wt-Cre:* GTAGGTGGAAATTCTAGCATCATCC*Forward Cre*: GCGGTCTGGCAGTAAAAACTATC*Reverse Cre*: GTGAAACAGCATTGCTGTCACTT

### 4.4. Metabolism-Related Studies

Six-week-old cohorts of male control and KO mice were fed a standard chow diet (LFD) (Teklad 2018 global 18% protein; Envigo, Indianapolis, IN, USA) or a high-fat diet (HFD) (Teklad TD.06415 with 40%–45% of calories from fat; Envigo) for 16 weeks. Body weight increases and energy intake were calculated by weighing animals and food every week, assuming that the energy density for LFD and HFD are 13 and 19.8 kJ/g, respectively. For glucose tolerance tests, glucose in aqueous solution was administered intraperitoneally (2.5 g glucose/kg) to 6 h-starved mice, and glycaemia in blood obtained from the tail was measured 15, 30, 45, 60, 90, 120 and 150 min after glucose injection. For insulin tolerance tests, insulin (Actrapid; Novo Nordisk Pharma A/S, Bagsvaerd, Denmark) in saline solution was administered intraperitoneally (0.75 UI/kg) to mice, and glycaemia in blood obtained from the tail was measured 15, 30, 45 and 60 min after glucose injection.

After killing mice by decapitation, blood was collected in heparinized tubes for preparation of plasma, and liver, heart, subcutaneous (inguinal depot) and visceral (epididymal depot) white adipose tissue, and interscapular brown adipose tissue were removed, weighed, immediately frozen in liquid nitrogen and stored at −80 °C until processing. One leg was digested in 1 M NaOH overnight to isolate tibia. Tibia length was measured and used to normalize animal and tissue weights. Alanine aminotrasferase (ALT) and alkaline phosphatase (ALP) activities were assayed in plasma using the commercially available kits ALT Activity Assay (MAK052-1KT; Sigma-Aldrich, ST. Louis, MO, USA) and ALP Assay Kit Colorimetric (ab83369; Abcam Plc, Cambridge, UK), respectively. Aspartate aminotransferase (AST) and lactate dehydrogenase (LDH) activities were measured in serum using a C311 Cobas Analyzer (Roche Diagnostics GmbH, Mannheim, Germany) following manufacturer’s instructions. For HOMA-IR index calculation ((Glucose(mg/dL) × Insulin(mU/L))/405), plasma glucose and insulin levels from overnight- fasted mice were determined using an Accutrend system (Roche Diagnostics) and a commercially available ELISA kit (RSHAKRIN031R; BioVendor R&D, Brno, Czech Republic), respectively. For the tunicamycin-induced hepatosteatosis study, three-months-old male mice were intraperitoneally injected with saline (Control) or 2 mg/kg tunicamycin for 24 h under fasting conditions (*n* = 6/treatment).

### 4.5. DEN-Induced Hepatocarcinogenesis

Cohorts of male control and KO mice received a single intraperitoneal injection of diethylnitrosamine (DEN) (Sigma) (25 mg/kg) at postnatal day 15 to induce ACC. At six weeks of age, fA3/Cre- and KO mice were separated into two dietary groups and fed on either LFD or HFD for 24 weeks, when tumor analysis and final tissue collection were performed. Body weight and energy intake were calculated as described in the metabolic study. Two hours before sacrifice, mice were intraperitoneally injected with 50 mg/kg BrdU in saline. Mice were sacrificed and tissues collected as described in the metabolic study. Tumors in each liver lobe were counted and measured with a caliper. The large lobe of each liver was used for histological analysis. The remaining liver was divided into non-tumor-involved and tumor-involved tissue, snap-frozen in liquid nitrogen and stored at −80 °C until further biochemical analysis.

### 4.6. Mouse Partial Hepatectomy and Acute CCl_4_ Administration

Partial hepatectomies in mice (66% of liver mass) were performed as previously described [46]. For CCl_4_ administration mice received a single intraperitoneal injection of CCl_4_ (Sigma-Aldrich) (1 μL/g of body weight in olive oil, final volume 50 μL) as previously described [47]. Controls received the same volume of vehicle (olive oil).

### 4.7. Mouse Primary Hepatocytes

Mouse hepatocytes were isolated by liver collagenase perfusion and primary cultures were established essentially as previously described [48].

### 4.8. Histology and Immunohistochemistry

Liver and adipose tissues were embedded in Tissue-Tek OCT compound (Jung Tissue Freezing Medium) or fixed 4% paraformaldehyde and stored in 70% ethanol until further processing for frozen and paraffin block preparation, respectively. Frozen tissue sections were cut into 5 µm sections and stained with oil-red O (ORO) for lipid detection (Sigma—O0625). Paraffin-embedded tissue sections (3 µm in thickness) were air dried and further dried at 60 °C overnight and used for H&E and immunohistochemical staining. Immunohistochemistry was performed using an Autostainer Plus (Dako—Agilent) or manually. Prior to immunohistochemistry, sections were dewaxed and therefore epitope retrieval was performed using a PT Link (Dako, Agilent), autoclave or with proteinase K (S3020, Dako, Agilent). Washings were performed using the Wash Solution AR (Dako—AR10211-2). Quenching of endogenous peroxidase was performed by 10 min of incubation with Peroxidase-Blocking Solution (Dako REAL S2023). Unspecific unions were blocked using 10% of goat normal serum (Life Technologies 16210064) plus 5% BSA (Sigma, 10735078001) for 60 min. The commercial primary and secondary antibodies used were: BrdU (Purified Mouse Anti-BrdU Clone B44—347580, BD Bioscience); Ki67 (Rabbit polyclonal to Ki67—ab15580, Abcam); F4/80 (F4/80 Monoclonal Antibody (BM8)—14-4801-85, eBioscience™); Caspase 3 (Cleaved Caspase-3 (Asp175)—9661-S, Cell Signaling); Β-catenin (β-catenin rabbit polyclonal IgG (H-102)—sc-7199, Santa Cruz Biotechnology, Dallas, TX, USA); COXIV (Anti-OxPhos Complex IV Subunit I Mouse IgG2a, k—ref: 459600, Invitrogen); ready to use HRP goat anti-rabbit (InmunoLogic DPVR-110HRP); Biotin-SP (long spacer) AffiniPure Donkey Anti-Rat IgG(H + L) (Jackson Immunoresearch, 712-065-150). Antigen–antibody complexes were reveled with 3-3′-diaminobenzidine (K3468, Dako), with the same time exposure (1 min). Sections were counterstained with hematoxylin (Dako, S202084) and mounted with Mounting Medium, Toluene-Free (CS705, Dako) using a DakoCoverStainer. Specificity of staining was confirmed by omission of the primary antibody, and/or the corresponding isotype control (Mouse IgGIsotype Control, Abcam, ab37355 or Rabbit IgGIsotype Control, Abcam, ab27478). Slide images were captured using a Nanozoomer slide scanner (Hamamatsu) and analyzed using Image J and QuPath software. The analysis of tumor tissue includes 1–2 mm tumors.

### 4.9. Cell Lines and Culture Conditions

All HCC cell lines (Hep3B, HepG2, SNU423 and SNU182) were purchased from the American Type Culture Collection (ATCC) and were maintained in DMEM or RPMI-1640 medium supplemented with 10% FBS, 100 U/mL penicillin and 100 μg/mL streptomycin (Gibco, Life Technologies). HCC cell lines were cultured at 37 °C in a humidified atmosphere containing 5% carbon dioxide.

### 4.10. Transfection of siRNAs

Silencer pre-designed siRNAs targeting ARMCX3 (5′-GGCUUAAAGUAUACAUGAATT-3′) and Scrambled siRNA (negative control), were purchased from Ambion (ThermoFisher, Waltham, MA USA). Hep3B, SNU423 and SNU182 cell lines were transfected with ARMCX3 and scrambled siRNA at a final concentration of 10 nM with Lipofectamine RNAiMAX reagent (ThermoFisher, Waltham, MA, USA) in Opti-MEM Medium according to the manufacturer’s instructions. After 5 h incubation, the media were replaced with fresh medium containing 10% FBS. After 24–48 h further incubation, cells were harvested for further analysis. 

### 4.11. Armcx3 Overexpression and SOX9 Knockdown In Vitro

Human hepatoma cell lines SNU423 or HepG2 were transduced with an adenoviral vector (Type 5 dE1/E3) containing GFP (transduction control) or the murine transgene Armcx3 (ARMCX3) (ADV-253053, Vector Biolabs, Malvern, PA, USA). Human and murine ARMCX3 proteins share a 97% of identity (Appendix A) according to BLASTP alignment [49]. Cells were harvested for further analysis after 48 h. In experiments of combined transduction and transfection, cells were transfected with control or *SOX9* siRNA (sc-36533; Santa Cruz Biotechnology) at a final concentration of 50 nM using Lipofectamine RNAiMAX reagent 6 h before adenoviral transduction.

### 4.12. Wound Healing Assay

Hep 3B, SNU423 cells seeded in 6-well plates at 80% of confluence were transfected with 10 nM siRNA as described above. Once the cells reached 90% confluence, a wound area was carefully created by scraping the cell monolayer with a sterile 10 μL pipette tip. The cells were then washed once with DPBS to remove detached cells. Subsequently, the cells were incubated at 37 °C in 5% carbon dioxide. The width of the wound area was monitored with an inverted microscope at various time points. 

### 4.13. Cell Viability Assay

The effect of siRNA on cell viability was measured using the MTT assay (Sigma) according to the manufacturer’s protocol. Hep3B, SNU423 and SNU182 cells (1000 cells/well) seeded in 96-well plates were transfected with 5 pmol siRNA as described above. After 24, 48 and 72 h of transfection, 100 μL of (4,5-dimethyl-2-thiazoyl)-2,5-diphenyl-2H-tetrazolium bromide (MTT) solution at 0.5 mg/mL (Sigma, St Louis, MO, USA) in fresh media was added to each well and were incubated at 37 °C in a 5% CO2-humidified incubator. Supernatants were discarded, and purple-colored precipitates of formazan were dissolved in 150 μL dimethyl sulfoxide by placing the plates on a low-speed shaker for 10 min to fully dissolve the crystals. Absorbance was measured at optical density (OD) of 590 nm using a microplate reader (BioTek, Winooski, VT, USA). The experiment was performed in triplicate wells and repeated three times.

### 4.14. Clonogenic Assay

After transfection for 24 h, HCC cells were seeded in six-well plates at 500 cells per well and incubated at 37 °C incubator for 10–15 days. Cell colonies were washed twice with PBS, fixed with 95% ethanol for 10 min, and stained with 0.1% crystal violet for 20 min. Plates were rinsed with water until water got colorless and dried. Colorant was solubilized with 10% acetic acid. Absorbance was measured at optical density (OD) of 570 nm using a microplate reader. Each experiment was performed in triplicate.

### 4.15. Palmitic Acid and Oleic Acid Treatments

HepG2 cells were treated with 0.33 mM palmitic acid or 0.66 mM oleic acid (Sigma-Aldrich) for 24 h. Fatty acids were dissolved in isopropanol and added to serum-free culture media containing 1% fatty acid free bovine serum albumin (BSA) to the desired final concentration as reported [50]. Controls received same volume of isopropanol (always <1% in the medium).

### 4.16. Western Blotting

Frozen liver tissue was homogenized in RIPA buffer (50 Mm TrisHCl pH 7.4, 150 mMNaCl, 1 mM EDTA, 1% *v*/*v* Triton X-100, 0.1% SDS), containing protease inhibitor cocktail (Roche Diagnostics) and phosphatase inhibitors (2 mM sodium orthovanadate, 1 mM sodium pyrophosphate, 10 mM sodium fluoride). Lysates were centrifuged at 16,000× *g* at ℃ for 10 min. Protein concentration was measured using the bicinchoninic acid (BCA) protein assay as specified by the manufacturer (Pierce, Thermo Fisher Scientific Inc; Rockford, IL, USA) and samples were stored at −80 °C until they were used. Samples were resolved by SDS-polyacrylamide gels and transferred onto nitrocellulose membranes. Membranes were blocked for 1 h at room temperature (RT) in TBST (Tris 10 mM (pH 7.4), sodium chloride 140 mM (TBS) with 0.1% Tween 20 containing 3% BSA. Primary antibodies were incubated overnight at 4 °C in TBST–0.02% azide. After incubation with HRP-labelled secondary antibodies for 1 h at RT in TBST–3% BSA, membranes were developed with the ECL system (GE Healthcare) and exposed to autoradiographic film (Kodak, Rochester, NY, USA); for quantification, images were acquired and quantified using ImageJ software. Primary and Secondary antibodies used were: rabbit anti-UCP1 1/1000 (ab10983; Abcam), mouse anti tubulin 1/1000 (T9026-100ul; Sigma-Aldrich), mouse anti pan-ERK 1/1000 (610123; Transduction Laboratories), rabbit Phospho-p44/42 MAPK (Erk1/2) (Thr202/Tyr204) 1/1000 (9101; Cell Signaling), rabbit p38 MAPK 1/1000 (9212; Cell-Signaling), rabbit Phospho-p38 MAPK (Thr180/Tyr182) 1/1000 (9211; Cell Signaling), rabbit active β-catenin 1/1000 (4270; Cell-Signaling), mouse total β-catenin 1/1000 (610154; BD Biosciences, San Jose, CA), goat anti-rabbit 1/3000 (ab6721; Abcam) and goat anti-mouse 1/3000 (Cat # 170-6516; Bio-Rad Laboratories, Hercules, CA, USA). Ponceau staining was also performed as protein loading control. For co-immunoprecipitation, cell lysates were incubated with rabbit IgG isotype control (ab172730, Abcam) or anti-SOX9 antibody 1/60 (ab185230, Abcam) conjugated with SureBeads Protein A Magnetic Beads (Bio-Rad) for 2 h and eluted with Laemmli buffer. Samples were immunobloted with rabbit anti-ARMCX3 antibody 1/1000 (25705-1-AP; Proteintech, Rosemont, IL, USA).

### 4.17. RNA Isolation, cDNA Synthesis, and Real-Time PCR

Dissected tissues (Liver, BAT and iWAT) were homogenized using a TissueLyser LT (Qiagen, Germany). Total RNA from homogenized tissues or cells was isolated using a column affinity-based method (NucleoSpin RNA II; Macherey-Nagel, Düren, Germany). Total RNA (500 ng) was transcribed into cDNA using TaqMan Reverse Transcription Reagents (Applied Biosystems/Life Technologies, Foster City, CA, USA). For quantitative analysis of mRNA expression, TaqMan quantitative real-time polymerase chain reaction (qPCR) was performed on a 7500 Real-Time PCR System (Applied Biosystems) using the following specific primer pair/probe sets: mouse *Armcx3* (Mm01968549_s1) or human *ARMCX3* (Hs01879444_s1). Relative mRNA levels of target genes were normalized with respect to that of 18S rRNA (Hs99999901_s1) or *GAPDH* (Hs02786624_g1), using the comparative (2-∆CT) method. Transcript levels were considered undetectable in cases where the CT value was >40 under our experimental conditions.

### 4.18. Statistics

Statistical analyses were performed using GraphPad Prism software version 5.03 (GraphPad Software Inc., San Diego, CA, USA). T-student, Mann-Whitney or Two-Way ANOVA test with Tukey or Dunnett post hoc corrections were applied. Correlation was established based on linear regression analysis. Differences at *p* ≤0.05 were considered significant. * = *p* ≤0.05, ** = *p* ≤0.01, *** = *p* ≤0.001, **** = *p* ≤0.0001.

## 5. Conclusions

In conclusion, we herein report that ARMCX3 contributes to hepatocyte proliferation and HCC progression, particularly in response to metabolic insults associated with lipotoxicity and through cellular mechanisms involving SOX9 interaction (Figure 8). These findings attribute a totally novel role to the biological function of the ARMCX3 member of ARMCX protein family. Along with our observation of high *ARMCX3* mRNA levels in a cohort of patients with HCC, our findings indicating that ARMCX3 promotes HCC suggest that inhibitory molecules targeting ARMCX3 may be of interest for the control of tumorigenic activity in liver.

## Figures and Tables

**Figure 1 cancers-13-01110-f001:**
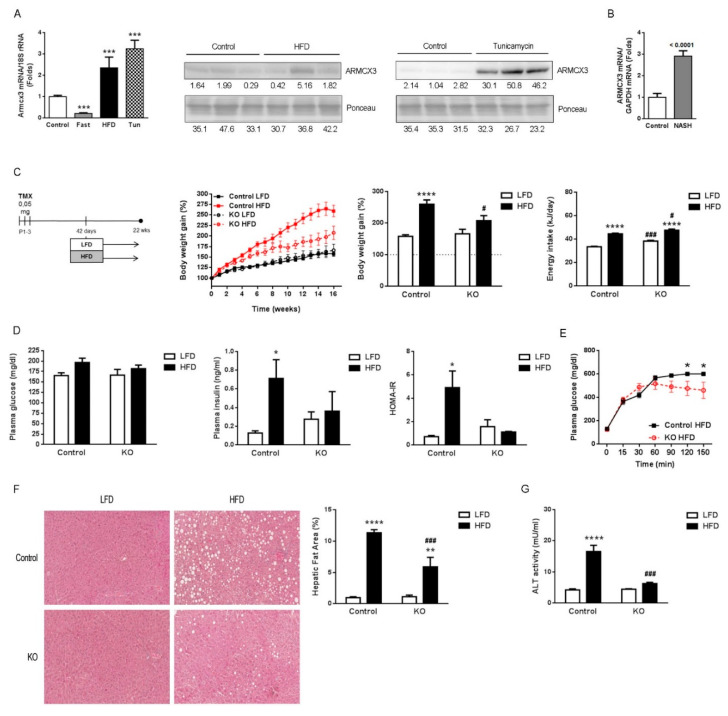
ARMCX3 deletion reduces the susceptibility of mice to high-fat diet (HFD)-induced obesity and metabolic alterations. (**A**) *Armcx3* mRNA levels in livers of mice exposed to regular feeding conditions (Control), overnight fasting (Fast), high-fat diet (HFD) or 24 h after intraperitoneal injection with 2 mg/kg tunicamycin (Tun) (left). Immunoblot of ARMCX3 protein levels and loading control (Ponceau staining) in livers from mice fed HFD or treated with tunicamycin for 24 h (right) (For uncropped immunoblots here and thereafter, see File S1). (**B**) *ARMCX3* mRNA levels in liver from healthy individuals (control, *n* = 19) and NASH patients (*n* = 24). (**C**) Schematic representation of the study design: 6-week-old *f*Armcx3/Cre- (Control) and ARMCX3-KO mice (tamoxifen-induced Cre activity) (KO) were fed low-fat diet (LFD) or HFD for 16 weeks, (left). Body weight curves (left, central), body weight gain (right, central) and dietary energy intake (right). (**D**) Glycemia (left), insulinemia (middle) and HOMA-IR index (right). (**E**) Glucose tolerance curves. (**F**) Representative microscopic pictures of H&E-stained livers (left), and quantification of the percentage of total area occupied by fat (right). Magnification: 10×. (**G**) Serum ALT levels. Bars indicate means ± SEM; *n* = 6–8 mice per group. T-student, Mann-Whitney or Two-Way ANOVA test with Tukey or Dunnet post hoc corrections were used. * *p* ≤ 0.05, ** *p* ≤ 0.01, *** *p* ≤ 0.001 and **** *p* ≤ 0.0001 for comparisons between diets or treatments, and ^#^
*p* ≤ 0.05, ^###^
*p* ≤ 0.001 for comparisons between genotypes.

**Figure 2 cancers-13-01110-f002:**
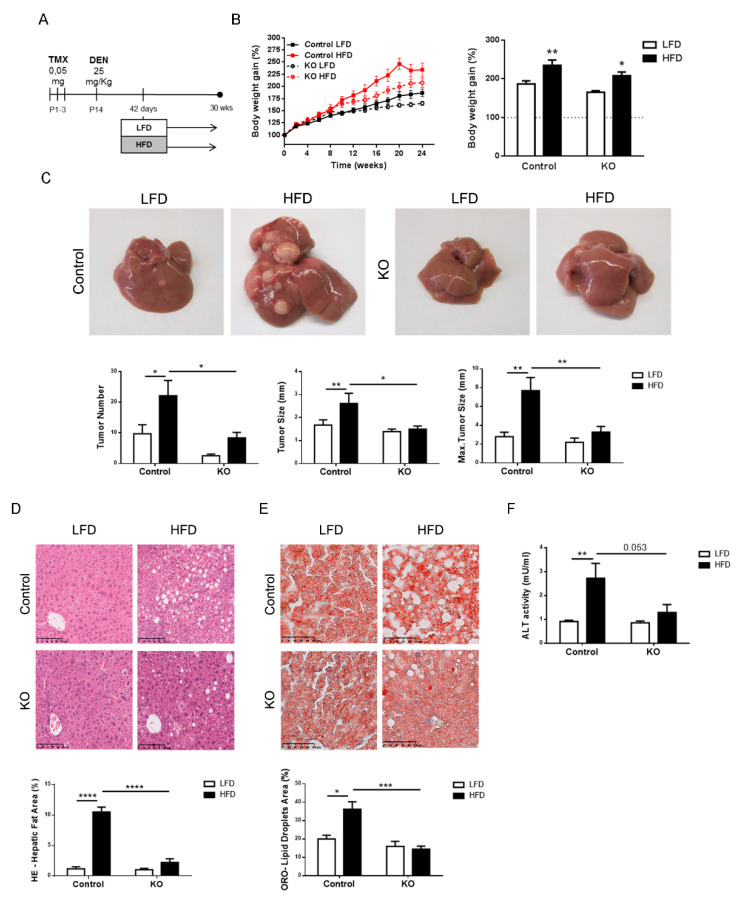
ARMCX3 deletion decreases diethylnitrosamine (DEN)-induced tumorigenesis and steatohepatitis in mice exposed to HFD. Data were obtained from 7-month-old *f*Armcx3/Cre- (Control) or ARMCX3-KO mice (KO) treated with DEN and maintained on LFD or HFD. (**A**) Schematic representation of the study design: 15-day-old *f*Armcx3/Cre- (Control) and ARMCX3-KO mice were injected with 25 mg/kg DEN and, beginning at week 6, fed LFD or HFD for 24 weeks. (**B**) Body weight curves (left) and body weight gain (right). (**C**) Representative pictures of livers (top) and quantifications of tumor number, size and maximal tumor size (bottom). Representative images and lipid accumulation quantification in liver sections stained with hematoxylin-eosin (**D**) and Oil-Red O (**E**). Scale bar: 100 µm. (**F**) Serum ALT levels. Bars show means ± SEM; *n* = 9–13 mice per group. Two-Way ANOVA test with Tukey post hoc correction was used. * *p* ≤ 0.05, ** *p* ≤ 0.01, *** *p* ≤ 0.001 and **** *p* ≤ 0.0001 for comparisons between diets or treatments.

**Figure 3 cancers-13-01110-f003:**
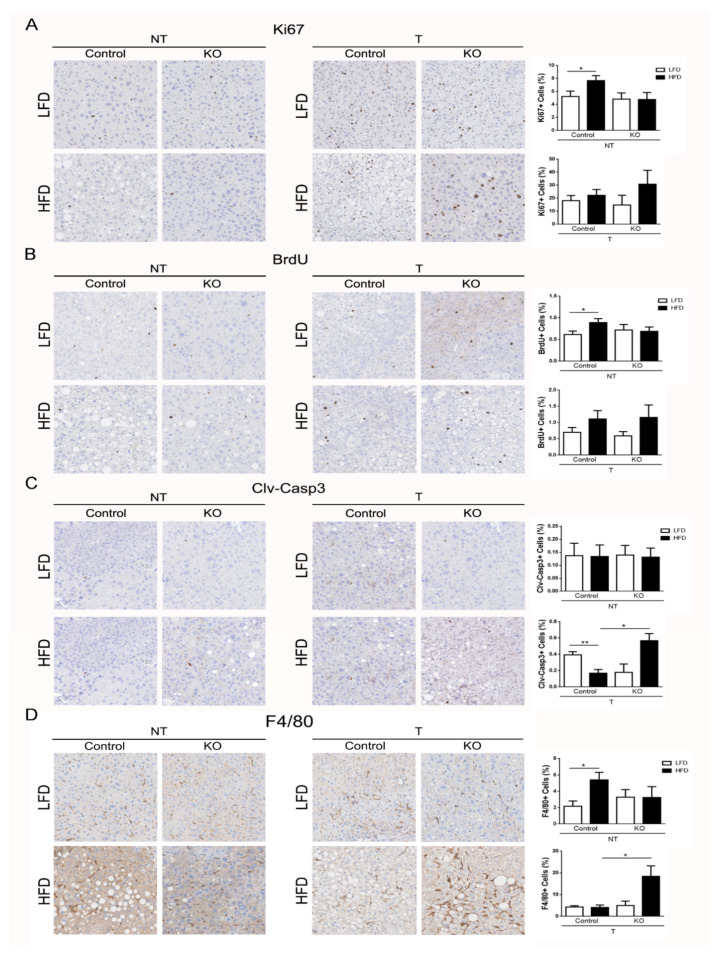
ARMCX3 deletion increases apoptosis and macrophage infiltration in hepatic tumors from HFD-fed mice. Data correspond to liver sections from 7-month-old *f*A3/Cre- (Control) or ARMCX3-KO mice treated with DEN and maintained on LFD or HFD. Representative liver sections from non-tumor (NT) and tumor (T) sections (left) and quantifications (right) of immunostaining for Ki67 (**A**), BrdU (**B**), cleaved caspase-3 (**C**) and F4/80 (**D**). Data are means ± SEM of 2–8 mice. Two-Way ANOVA test with Tukey post hoc correction was used. * *p* ≤ 0.05 and ** *p* ≤ 0.01. Magnification: 13×.

**Figure 4 cancers-13-01110-f004:**
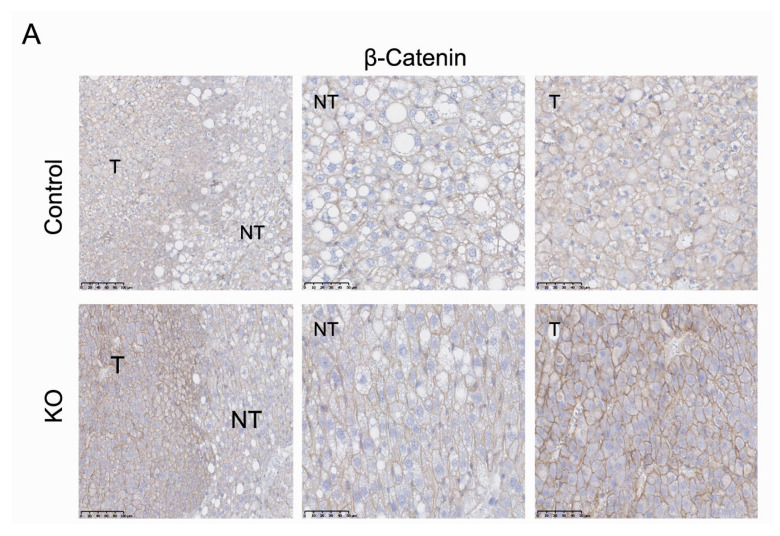
ARMCX3 deletion affects ERK signaling in hepatic tumors from HFD-fed mice. (**A**) Representative staining of β-catenin in liver sections from Control or ARMCX3-KO mice maintained on HFD. Sections include both non-tumor (NT) and tumor (T) regions and magnifications are shown. Scale bars: 100 µm and 50 µm (magnified). (**B**) Immunoblot analysis of the phosphorylation of ERK and p38 (representing pathway activation) in control and ARMCX3-KO mice exposed to HFD. Shown are representative immunoblot images (left) and quantification of phosphorylated/total ratios for ERK and p38 (right). Bars indicate means ± SEM; *n* = 2–4 mice per group. Two-Way ANOVA test with Tukey post hoc correction was used. * *p* ≤ 0.05 for comparisons between Control and KO samples.

**Figure 5 cancers-13-01110-f005:**
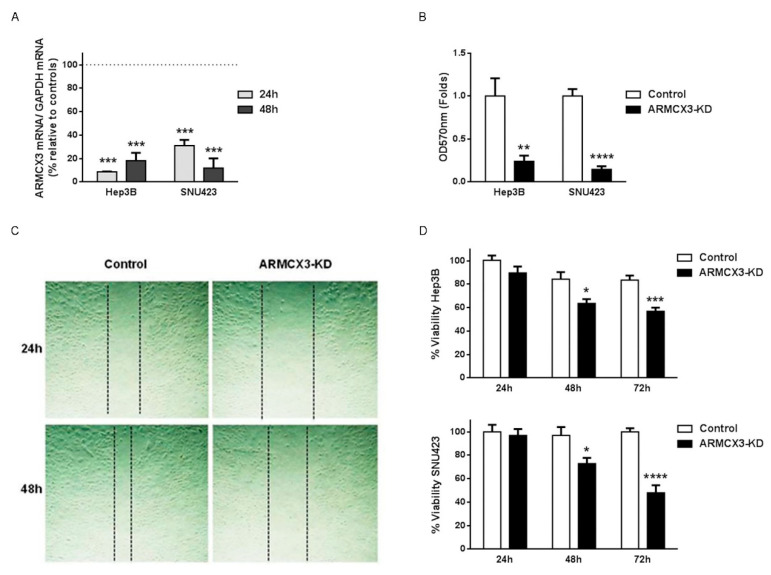
ARMCX3 deletion reduces the viability, clonality and invasivity of HCC cells in vitro. (**A**) ARMCX3 mRNA levels in Hep3B and SNU423 HCC cells 24 h and 48 h after transfection with ARMCX3 siRNA versus negative control siRNA (top dotted line), **** *p* ≤ 0.0001. (**B**) Colony-forming assays performed with Hep3B and SNU423 cells transfected with negative control siRNA (Control) or ARMCX3 siRNA. (**C**) Representative images of wound-healing assay results obtained using control and ARMCX3-KD Hep3B cells. Magnification: 10× (**D**) Percentage viability of control and ARMCX3-KD Hep3B and SNU423 cells at 24 h, 48 h and 72 h after transfection. Bars show the means ± SEM of three independent cell culture points. Two-Way ANOVA test with Dunnett post hoc correction was used. * *p* ≤ 0.05, ** *p* ≤ 0.01, *** *p* ≤ 0.001 and **** *p* ≤ 0.0001.

**Figure 6 cancers-13-01110-f006:**
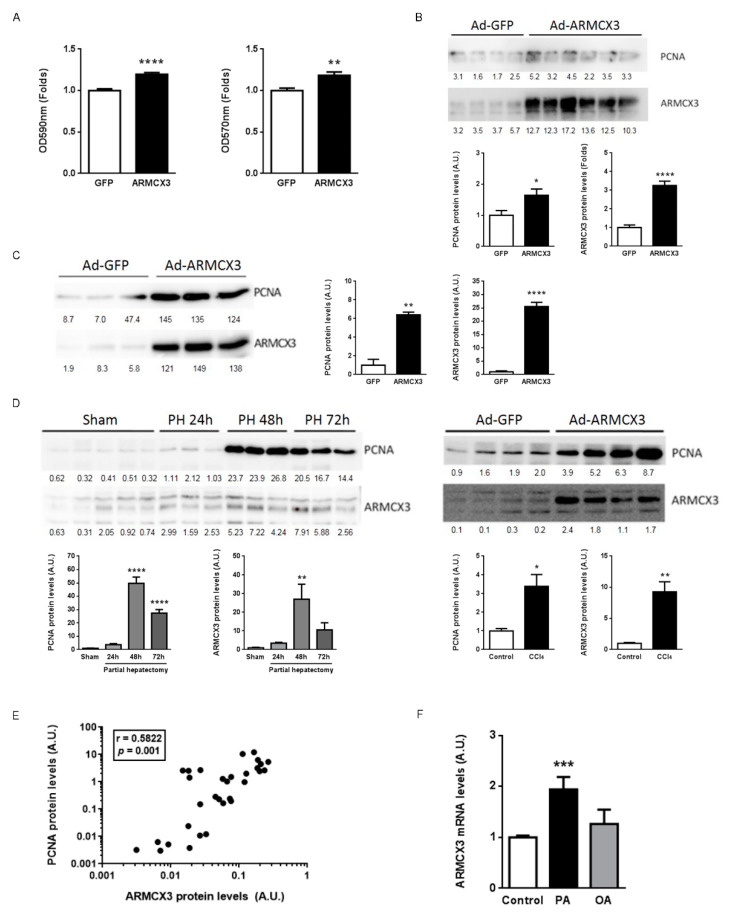
ARMCX3 expression correlates with the proliferative capacity in hepatocytes. (**A**) MTT and colony-forming assays in SNU423 cells overexpressing ARMCX3 at 48 h post-transduction. (**B**) PCNA protein levels in the cells described in (**A**), presented as representative immunoblots (top) and quantification (bottom) of PCNA and ARMCX3 levels. (**C**) PCNA protein levels in primary hepatocytes overexpressing ARMCX3 at 48 h post-transduction, presented as representative immunoblots (left) and quantification (right) of PCNA and ARMCX3 levels. (**D**) PCNA and ARMCX3 protein levels in livers of mice subjected to partial hepatectomy at 24 h, 48 h or 72 h post-surgery, and in livers of mice treated with CCl_4_ for 48 h; representative immunoblots (top) and quantifications (bottom) are presented. (**E**) Pearson’s correlation between ARMCX3 and PCNA protein levels in hepatocytes (*n* = 30). (**F**) ARMCX3 mRNA levels in HepG2 cells treated with palmitic acid (0.3 mM) or oleic acid (0.6 mM) for 24 h. Ponceau staining, which was used as a loading control in the immunoblots, is shown in Appendix A. T-student or Mann-Whitney test with Dunnett post hoc corrections were used. * *p* ≤ 0.05, ** *p* ≤ 0.01, *** *p* ≤ 0.001 and **** *p* ≤ 0.0001.

**Figure 7 cancers-13-01110-f007:**
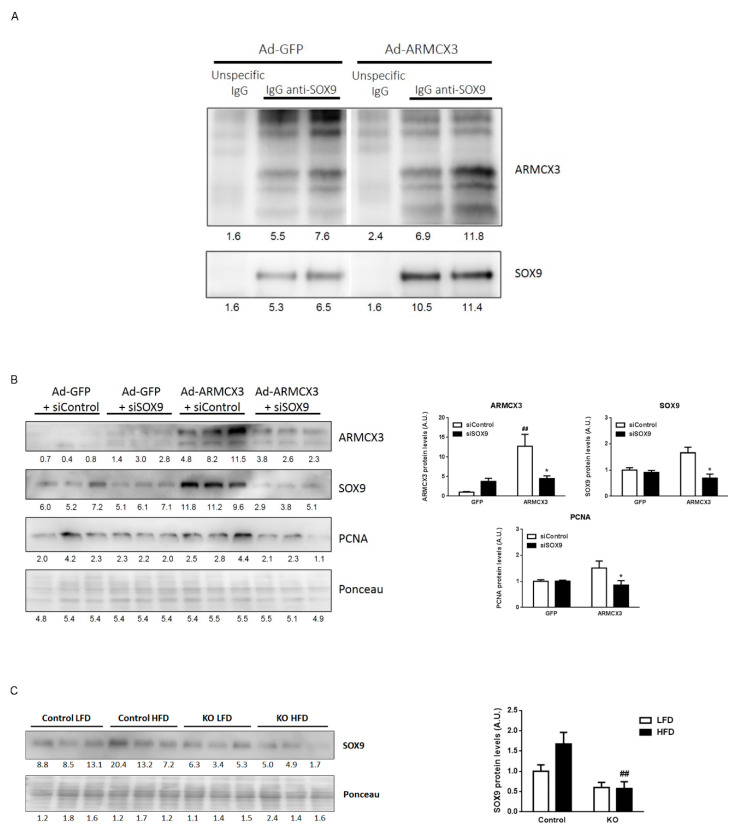
ARMCX3-induced hepatocyte proliferation involves SOX9. (**A**) Immunodetection of ARMCX3 and SOX9 in SOX9 pull-down samples from HepG2 cells transduced with Ad-GFP or Ad-ARMCX3 at 48 h post-transduction. (**B**) ARMCX3, SOX9 and PCNA levels in HepG2 cells at 48 h after cells were transduced with Ad-GFP or Ad-ARMCX3 and transfected with scrambled siRNA (Control) or SOX9 siRNA. Representative immunoblots (top) and quantification (bottom) are shown. Bars indicate means ± SEM of three independent experiments. Two-Way ANOVA test with Tukey post hoc correction was used. * *p* ≤ 0.05 for comparisons between siControl and siSOX9, and ^##^
*p* ≤ 0.01 for comparisons between GFP and ARMCX3. (**C**) SOX9 levels in 7-month-old fArmcx3/Cre- (Control) or ARMCX3-KO mice (KO) treated with DEN and maintained on LFD or HFD for 24 weeks. Bars indicate means ± SEM; *n* = 9–13 mice per group. Two-Way ANOVA test with Tukey post hoc correction was used. ^##^
*p* ≤ 0.01 for comparisons between genotypes.

**Figure 8 cancers-13-01110-f008:**
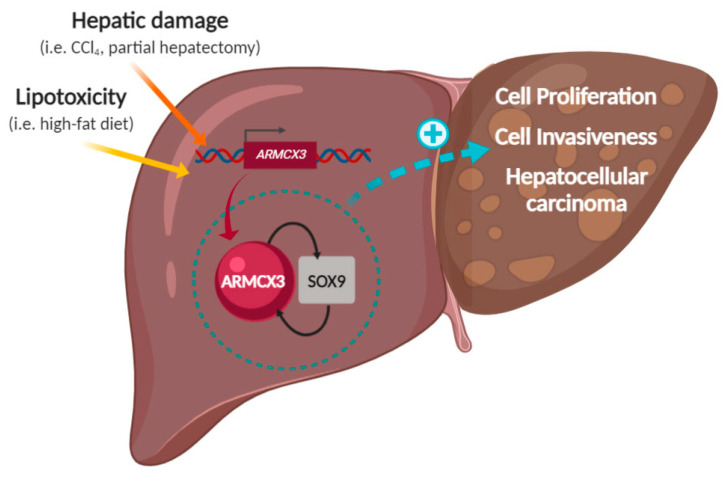
Schematic representation of the role of hepatic ARMCX3, in interaction with SOX9, in eliciting proliferative signals and hepatocarcinogenesis in response to lipotoxicity and hepatic damaging signals.

## Data Availability

The data presented in this study are available in this article (and Appendix A).

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
