# Peer review of "ARMCX3 Mediates Susceptibility to Hepatic Tumorigenesis Promoted by Dietary Lipotoxicity"

_cancers, 2021, doi:10.3390/cancers13051110_

Round 1

Reviewer 1 Report

I suggest this manuscript can be published now.

Reviewer 2 Report

Congratulations to the authors for the new version of the manuscript, where they have resolved all the comments of the reviewers, and that it must be accepted in the current version.

This manuscript is a resubmission of an earlier submission. The following is a list of the peer review reports and author responses from that submission.

Round 1

Reviewer 1 Report

In the manuscript entitled" ARMCX3 mediates susceptibility to hepatic 2 tumorigenesis promoted by dietary lipotoxicity" (cancers-987859), the author showed Armcx3 is strongly increased in mouse liver in response to lipid availability and proliferation-inducing malignancy. In patients, the levels of hepatic Armcx3 are also increased in conditions of high exposure of the liver to fat. Finally, forced Armcx3 was found to promote hepatoma cell proliferation through the interaction with Sox9, a known proliferation factor in hepatocellular carcinoma.

ARMCX3(ALEX3), a member of tumor suppressor family, is a mitochondria protein. Evidence had showed ALEX3 have tumor suppression effects on lung and renal cancer via SOXs protein and Wnts signals (Tumour Biol. 2017 Jul;39(7): 1010428317701441.) (PLoS ONE 8:E67773-E67773(2013)) (J Biol Chem, 2009 May 15.). The effect of Alex3 on inhibiting invasion and migration may attribute to upregulation of E-cadherin expression through AKT-Slug pathway inactivation. Accordingly, based on published manuscript, the study has lower novelty on this mechanism exploring on this study filed and don’t suitable for publishing in cancers. Further, many data set is non-consistent to each other from a chimera animal model mixed with chemical injury and then HFD and in vitro system providing confused evidence in many Figures. I think it is a poor organized manuscript.

Author Reply:

While we appreciate the time and effort Reviewer 1 put into their comments, we must respectfully disagree with several of his/her main criticisms of our work.
First, we strongly believe that Reviewer 1 misunderstood our main conclusions, leading to their misapprehension that our work lacked novelty. We are fully aware of the research reported in the three articles that the reviewer mentioned, and indeed included them in our submitted manuscript (Refs 9, 5 and 24, in the submitted manuscript). These previous studies found that Armcx3 appears to act as a tumor-suppressor in lung- and kidney-based systems in vitro. In complete contrast, our manuscript reports that Armxc3 promotes tumorigenesis in a hepatic system, and that its loss-of-function has protective (i.e., tumor suppressive) effects. In this sense, we think that the novelty is obvious (different system, opposite results).
We also believe that there was some misunderstanding by Reviewer 1 regarding the effects of Armcx3 (Alex3) on invasion and migration. We do not report the “effects of Alex3 inhibiting invasion and migration”, but instead show the opposite effect. For example, a sentence in the abstract reads as follows: “ARMCX3 downregulation reduced the viability, clonality and migration of HCC cell lines, whereas ARMCX3 overexpression caused the reciprocal effects”. Thus, our findings are not redundant with respect to the
data previously reported in the mentioned articles, but rather are discrepant with respect to the prior work (and arise from cells of a different origin). In the revised manuscript, we clarified this further by adding a final conclusive sentence, as well as by explicitly indicating that ARMCX3 promotes tumorigenesis (by using a “plus” symbol) in liver in the final schematic depiction of our results.
Regarding the clarity and organization of the data in the manuscript, we have revised some statements to further explain our conclusions.
Finally, we regret that Reviewer 1 was confused by our use of several different models (the loss-of-function in vivo model and the siRNA-mediated loss-of-function and overexpression experiments in cell culture models). We do not identify major inconsistencies in the conclusions obtained using animal- and cell-based experiments (apart from expectable minor differences due to the distinct models used) and, in fact, we believe that our concordant indications that ALEX3 favors hepatic tumorigenesis, obtained in the distinct models, actually added strength to the story. Regarding the utilized mouse model, in which the application of DEN and HFD allowed us to explore lipotoxicity-promoted hepatic tumorigenesis, we would like to note that this is a standard experimental model that has been commonly reported in the literature (see, for example, the manuscript in Cell by M.Karin and collaborators, Cell 2010;140:197-208., Ref 20 in the original manuscript, among many others).
In light of the points raised above, we respectfully ask that this reviewer’s response to our manuscript be carefully re-assessed in relation to the novelty of our work and the reliability of our experimental approaches.

Reviewer 2 Report

In this manuscript, Mirra and colleagues report that ARMCX3 is a novel molecular modifier in liver carcinogenesis. ARMCX3 downregulation inhibited liver tumor growth both in vivo and in vitro, which is mainly mediated by SOX9.

Comments

  1. The author demonstrated ARMCX3-KO mice did not display any obvious behavioral defect and protected against NAFLD. Besides liver fat level and Serum ALT, LFTs should be involved in here to answer above two issues.
  2. The author proved that ARMCX3 affects ERK signaling in DEN-induced tumors from HFD-fed mice,β-catenin, p38 and ERK signaling tested in DEN, HFD-mice should also be detected in your in vitro experiments.
  3. It’s interesting that ARMCX3 promotes the expression of PCNA and SOX9. The correlation status of ARMCX3 and PCNA or SOX9 should be checked in TCGA database as well as your cohort of 48 HCC patients. And the level of SOX9 in your DEN mice model should be added.

Author Reply:

We performed an expanded analysis of LFTs (AST, ALP, LDH) in blood from ARMCX3-KO models. Although AST levels tended to exhibit small decreases in the ARMCX3-KO models, the effects were not as marked as those for ALT. The relevant data are shown
in Supplemental Tables 3 and 4. Accordingly, the results and conclusions regarding NAFLD have been toned down in the revised manuscript (see pag9, end of section 2.2.).

1.In accordance with the reviewer’s suggestion, we determined the phosphorylation statuses of β-catenin, p38 and ERK in in-vitro-cultured hepatocytes overexpressing ARMCX3. The levels of P-ERK/ERK were increased in this experimental setting, which is concordant with the reciprocal observation (decreased levels) seen in livers from ARMCX3-KO mice in vivo. We further found that β-catenin and p38 tended to increase in ARMCX3-overexpressing hepatic cells in vitro, whereas no such change had been observed in our ARMCX3-KO model mice in vivo. These results are now presented in the revised manuscript and shown in Supplemental Figure S6. The more marked effect of ARMCX3 manipulation in the in vitro overexpression model compared to the in vivo knockout model may reflect that the former is a cell-based acute and transient ARMCX3 intervention model whereas the latter is a whole-animal-based sustained intervention model that may be more able to be associated with homeostatic mechanisms to compensate partially for alteration of ARMCX3. Regardless, the concordant results obtained from the two model types for our analyses of the ERK pathway suggest that this pathway is the most consistently identified as mediator of the intracellular actions of ARMCX3.

2.We performed the suggested correlation analysis. Our analyses of data from the TCGA database and our HCC cohort consistently failed to find a significant correlation between the transcript level of ARMCX3 or those of PCNA or SOX9. Of note, the strong correlation between ARMCX3 and PCNA levels shown in Figure 6E reflects protein expression levels in our animal and cellular models whereas available data in TCGA and our HCC cohort correspond to mRNA levels. Perhaps this may explain the lack of correlation using the databases above. Alternatively (or in addition), human patients in vivo may experience more moderate alterations of ARMCX3 expression than those seen in the model systems, which are subjected to more dramatic changes (full invalidation in vivo, forced over-expression or invalidation in vitro), which may have precluded us from detecting a significant correlation in TDBA and our HCC databases.
As suggested by the reviewer, we analyzed SOX9 levels in the mouse models and found that ARMCX3 depletion in liver leads to a concomitant reduction in SOX9 protein levels
(Fig. 7 in the revised manuscript). This is a reciprocal finding relative to the results obtained with ARMCX3 overexpression in vitro, and is fully concordant with our proposal that there is regulatory interdependence between ARMCX3 and SOX9.

Reviewer 3 Report

Congratulations to the authors for the fantastic manuscript that I have had the pleasure to review, with a lot of work in which they have used a KO mouse model for ARMCX3, verified in patient samples, in cell cultures, ... a work with a run and incredible coherence, which will only need a minor revision with small contributions and suggestions, which the authors can include in the new version.

The introduction should probably be expanded a bit.

Some abbreviations should also be detailed the first time they appear, such as DEN (diethylnitrosamine) that the authors use to induce HCC, and perhaps this reference DOI: 10.3892 / etm_00000062 could be included, where the authors find differences in epithelial cancer in ARMCX1 and ARMCX2 as indicated in the introduction for various types of cancer, and where the authors also suggest quite consistently that ARMCX6 may be a novel biomarker of cell fate determination in specific tissues or neoplasms derived from certain tissues.

Regarding the 2D figure, although these are personal preferences, I prefer to put the centrilobular vein in a corner of the picture and not in the center, since it allows a greater perimeter view, especially when they are small images.

Finally, congratulate the authors for the interesting find and the fantastic work done.

Author Reply:

Congratulations to the authors for the fantastic manuscript that I have had the pleasure to review, with a lot of work in which they have used a KO mouse model for ARMCX3, verified in patient samples, in cell cultures, ... a work with a run and incredible coherence, which will only need a minor revision with small contributions and suggestions, which the authors can include in the new version.
The introduction should probably be expanded a bit.
We are grateful to the reviewer for their positive appreciation of our work. As suggested, we have expanded the Introduction, such as by including more details regarding the roles previously reported for members of the Armcx3 family in tumorigenesis.
Some abbreviations should also be detailed the first time they appear, such as DEN (diethylnitrosamine) that the authors use to induce HCC, and perhaps this reference DOI: 10.3892 / etm_00000062 could be included, where the authors find differences in epithelial cancer in ARMCX1 and ARMCX2 as indicated in the introduction for various types of cancer, and where the authors also suggest quite consistently that ARMCX6 may be a novel biomarker of cell fate determination in specific tissues or neoplasms derived from certain tissues.
As suggested, we now provide full terms for each abbreviation as it is introduced in the text (e.g., DEN). We have also newly included the mentioned reference and expanded our discussion of the distinct roles played by members of the Armcx3 family in relation to tumorigenesis-related events, especially in relation to the type of Armcx member and the type of cellular system being targeted.
Regarding the 2D figure, although these are personal preferences, I prefer to put the centrilobular vein in a corner of the picture and not in the center, since it allows a greater perimeter view, especially when they are small images.
We have replaced the representative images provided, as suggested by the reviewer.
Finally, congratulate the authors for the interesting find and the fantastic work done.
Thank you for your positive appreciation.